# To Repeat or Not To Repeat:
# Insights from Scaling LLM under Token-Crisis

**Fuzhao Xue**[1]    **Yao Fu**[2]    **Wangchunshu Zhou**[3]    **Zangwei Zheng**[1]    **Yang You**[1][†]

[1]National University of Singapore
[2]University of Edinburgh
[3]ETH Zurich

## Abstract

Recent research has highlighted the importance of dataset size in scaling language models. However, large language models (LLMs) are notoriously token-hungry during pre-training, and high-quality text data on the web is likely to be approaching its scaling limit for LLMs. To further enhance LLMs, a straightforward approach is to repeat the pre-training data for additional epochs. In this study, we empirically investigate three key aspects under this approach. First, we explore the consequences of repeating pre-training data, revealing that the model is susceptible to overfitting, leading to multi-epoch degradation. Second, we examine the key factors contributing to multi-epoch degradation, finding that significant factors include dataset size, model parameters, and training objectives, while less influential factors consist of dataset quality and model FLOPs. Finally, we explore whether widely used regularization can alleviate multi-epoch degradation. Most regularization techniques do not yield significant improvements, except for dropout, which demonstrates remarkable effectiveness but requires careful tuning when scaling up the model size. Additionally, we discover that leveraging mixture-of-experts (MoE) enables cost-effective and efficient hyper-parameter tuning for computationally intensive dense LLMs with comparable trainable parameters, potentially impacting efficient LLM development on a broader scale.

## 1 Introduction

Large Language Models (LLMs) have demonstrated remarkable performance on various NLP tasks [14, 21], and have even become a part of our daily lives through applications such as ChatGPT and Bard. This success has been largely attributed to scaling up transformer-based language models, as evidenced by recent work [10, 20, 28]. In the early stages of transformer scaling, researchers observed that larger models could achieve comparable performance with smaller models using fewer training steps and less pre-training data [10], leading to early views that model size might be one of the most critical factors in achieving better performance.

**Dataset size is more important than we thought in LLM scaling.** Recent work [8] found that the pre-training dataset size plays a more significant role than previously thought and proposed compute-optimal scaling (*i.e.,* Chinchilla scaling law), where model size and training dataset size should be scaled equally for optimal performance given a fixed computation budget. For instance, an under-trained larger model like Gopher-280B [20] can be outperformed by a well-trained smaller model like Chinchilla-70B if not enough data is used in larger model training. The intuition here is that the decreased model size can be compensated by the increased size of data. The effectiveness of the Chinchilla scaling law is further validated by the recent success of LLaMA-65B [31] and PaLM-2 [1].

---

[†]Correspondence to youy@comp.nus.edu.sg

37th Conference on Neural Information Processing Systems (NeurIPS 2023).

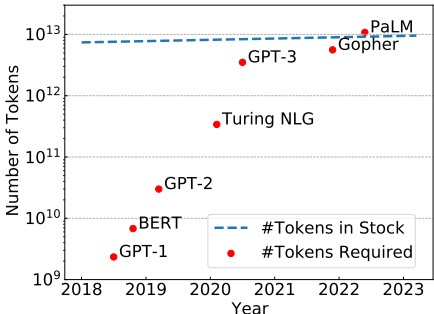

Figure 1: Modeling number of tokens in stock and number of tokens required by training compute-optimal LLM.

**Insufficient tokens hinder LLM scaling.** State-of-the-art LLMs require vast amounts of internet-scale text data for pre-training, such as the 780 billion tokens used for PaLM-540B [3] and the 1.4 trillion tokens used for Chinchilla-70B [8]. However, this raises two critical questions: (1) how many tokens are needed to fully train SoTA LLMs, and (2) how many tokens are available for pre-training? To answer these questions, we modeled token requirements using Chinchilla scaling law and estimated the scale of potential high-quality pre-training data based on recent research [32]. Unfortunately, as shown in Figure 1, the growth rate of high-quality text data on the internet is much slower than the growth rate of data required by LLMs. For instance, to fully pre-train PaLM-540B, 10.8 trillion tokens would be needed, but the total stock of high-quality text data is only around 9 trillion tokens. Moreover, the high-quality text data is growing at a rate of 4-5% per year, in line with the world economy, which is much slower than the pace of LLM growth and hardware improvement. According to recent study [32], high-quality text data may not suffice the requirements of scaling LLMs and in a pessimistic scenario, and we may run out of new data between 2023 and 2027. In light of the compute-optimal scaling study, this may already have occurred. Therefore, data may be becoming a more significant bottleneck for scaling transformers than hardware. In this paper, we refer to this problem as the "token-crisis".

In addition, it is important to note that the pre-training dataset size prediction discussed above is based on the compute-optimal training proposed in [8], which only considered the training cost of LLMs and ignored the inference cost. However, given that LLMs are often used as a service, such as in the case of Bard and Bing, and perform a significant amount of inference every day, it is crucial to consider the inference cost in compute-optimal modeling as well. Therefore, in order to achieve the best performance with the least computation cost per sample, it is likely that LLMs will require even more data than we estimated in Figure 1. This further emphasizes the importance of making full use of the off-the-shelf high-quality text data for LLM pre-training.

**The token-crisis is even more severe when it comes to non-English data.** According to the Web Technology Surveys[3], English content makes up over 56% of the web, with non-English data from over 100 languages comprising only 44% of the total. This long-tailed distribution of data makes it much harder for LLMs to perform well on non-English tasks. Despite PaLM's impressive 540B parameters and training on 780B tokens [3], including 22% non-English data, it still lags behind models such as mT5 [37] and ByT5 [36] on non-English tasks like Multilingual QA. With native English speakers making up just 5% of the world's population, achieving comparable performance on non-English tasks is highly desirable from fair access and democratizing LLMs perspectives.

**Using pre-training data repeatedly.** To alleviate the token-crisis, one straightforward approach is training LLM for multiple epochs. The practice of multi-epoch training varies in subfields of machine learning. Although there exist models like Vision Transformers [6] that are typically trained for many epochs (*e.g.,* 300 epochs on ImageNet [24]), LLMs are often trained for only one or a few epochs [3, 8, 31]. Currently, it is unclear what multiple epochs mean for language model pretraining: while Hoffmann *et al.* [8] suggest that training LLM with repeated tokens can be harmful to performance, Taylor *et al.* [30] observed improvements when training a 120B model for 4 epochs. Therefore, although training with repeated data may seem superficially simple, it has a nontrivial influence on practitioners and therefore further investigation is needed to determine its effects.

**Contributions and Insights** This paper presents a systematic empirical study of repeating pre-training data for token-crisis problem, making it the first work of its kind. We summarize 11 insights

---

[3]https://w3techs.com/technologies/overview/content_language

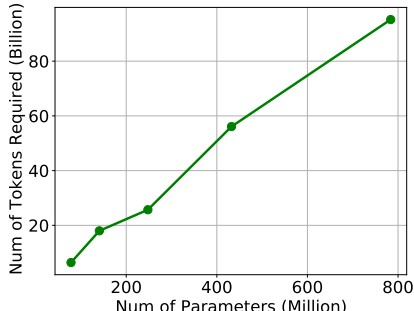

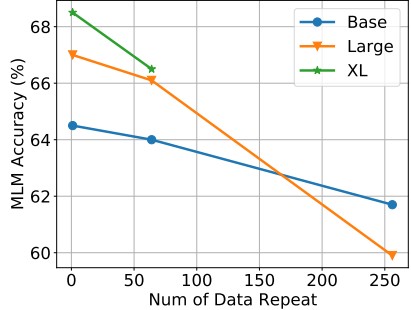

Figure 2: We observe a linear relationship between the number of tokens required to fully train the model and the number of model trainable parameters, which validates that the Chinchilla scaling law still holds when training T5 on C4 dataset.

Figure 3: We train models at three different scales (*i.e.,* T5-Base, T5-Large, T5-XL) with different dataset sizes but the same amount of total computation budget (*i.e.,* batch size 128, sequence length 512, training steps $2^{19}$ (around 524K).

in three aspects. First, we investigate what would happen when training with repeated pre-training data in Section 2 and found (1) Encoder-Decoder[4] model on C4 dataset is comparably data-hungry as stated in Chinchilla scaling law; (2) Larger models are more prone to overfitting when training with repeated data.

We study three components (*i.e.,* data, model, objectives) in Section 3 to explore the key factors contributing to multi-epoch degradation. We found (3) Training LLM with a larger dataset for multiple epochs can alleviate the multi-epoch degradation; (4) The use of high-quality data[5] does not mitigate multi-epoch degradation. (5) The number of parameters plays a crucial role in multi-epoch degradation, even when the computation budget is fixed. The effect of FLOPs on this issue is negligible; (6) The Mixture-of-Experts transformer can even be employed to predict the behavior of dense models that have comparable parameters but require much more computation; (7) Utilizing a mixture of training objectives, such as UL2 [29], can accelerate LLM learning, but it also leads to faster memorization, resulting in worse multi-epoch degradation.

We then investigate whether off-the-shelf regularization technologies can alleviate multi-epoch degradation in Section 4. We found (8) While most existing transformer regularization techniques struggle, dropout proves to be highly effective, despite its infrequent usage in LLM pre-training; (9) Dropout can be introduced only at the later stage of pre-training (after a few epochs) to ensure faster learning at the early stage of pre-training; (11) When scaling to very large models, dropout requires additional tuning.

We finally make use of insight (6) to alleviate the challenge introduced by insight (10). That is our final insight (11): The MoE model can serve as a more affordable and faster alternative to fine-tune hyper-parameters (e.g., dropout rate) of large dense models with comparable parameters but significantly more FLOPs. We think this approach has great potential to have a broader impact on efficient LLM development.

## 2 What Are the Consequences of Repeating Pre-training Data?

In this study, we adopt T5 1.1 as our default pre-training configuration. This means that, unless specified otherwise, we utilize the C4 dataset [21] along with the identical model architecture, training objectives, and hyper-parameters as described [21]. Detailed hyper-parameters can be found in Appendix J.

**Insight (1): Training T5 on C4 is Data-Hungry** In this section, our focus is to investigate the effects of training a LLM for multiple epochs under token-crisis conditions. Before delving into that, it is essential to examine whether we have a similar data-hungry observation like Chinchilla scaling

---

[4]We verified Encoder-Decoder is actually not that different from Decoder-only model in Appendix F

[5]The data quality here is relative. For instance, we think the quality of C4 is low when comparing with Wikipedia but C4 is good when comparing with C4 unclean. In addition, we limit the quality discussion here to web-scale pre-training data. High-quality instruction tuning data is not in this scope.

Table 1: We finetune pre-trained T5 checkpints on SQuAQ[22] dataset. The C4 Top-1 Acc denotes the masked token prediction accuracy on C4 validation set before fine-tuning. On SQuAD, we report both Exact Match (EM) and F1 score.

| Model | C4 | SQuAD | |
|---|---|---|---|
| | Val Acc | EM | F1 |
| T5 Base ($2^{35}$ tokens, 1 epoch) | 64.6 | 82.4 | 90.0 |
| T5 Base repeat $2^8$ ($2^{27}$ tokens, $2^8$ epochs) | 61.7 (-2.9) | 79.9 (-2.5) | 88.1 (-1.9) |

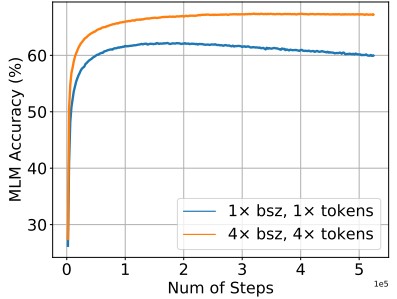

Figure 4: We repeatly use $2^{27}$ and $2^{29}$ tokens for $2^8$ times by using batch size 128 and 512 for $2^{18}$ steps.

Table 2: We pre-train two T5 models on a subset of C4 and a subset of Wikipedia with the same computation budget. Both of these two subsets have around *i.e.,* $2^{27}$ tokens. We then finetune these two checkpoints on SQuAD and report the Exact Match (EM) and F1 score.

| Dataset | SQuAD | |
|---|---|---|
| Acc | EM | F1 |
| C4 ($2^{35}$ tokens) | 82.4 | 90.0 |
| C4 ($2^{27}$ tokens) | 79.9 (-2.5) | 88.1 (-1.9) |
| Wikipedia ($2^{35}$ tokens) | 82.4 | 89.9 |
| Wikipedia ($2^{27}$ tokens) | 79.4 (-3.0) | 87.6 (-2.3) |

law in a widely-used open-sourced setting, specifically training an encoder-decoder transformer on the C4 dataset. To assess this, we follow [8] and train models with six different configurations. For detailed configurations, please refer to Appendix K. We then compare the validation accuracy for masked token prediction at various computation budgets. When a larger model outperforms a smaller model, it indicates that the smaller model has received sufficient tokens. The number of tokens used to train the smaller model can then be considered as the token requirement for full training. Figure 2 illustrates our findings, showing a linear relationship between the number of tokens required and the model size. Overall, our results indicate that the Chinchilla scaling law holds true when training T5 with the C4 dataset.

**Insight (2): Multi-epoch Degradation** As suggested in Figure 1, we anticipate encountering token scarcity issues as we continue to scale. Consequently, our next investigation revolves around training LLMs with repeated data. To explore this, we randomly select several subsets of the C4 dataset containing approximately $2^{35}$, $2^{29}$, and $2^{27}$ tokens, resulting in each token being repeated 1, $2^6$, and $2^8$ times, respectively. The results, as presented in Figure 3, demonstrate the expected performance degradation when training LLMs with repeated tokens. Furthermore, we observe that larger models are more susceptible to overfitting under token-crisis conditions. Specifically, when trained without a sufficiently large dataset, T5-XL, despite consuming more computational resources, performs worse than T5-Large having access to $4\times$ data ($2^{29}$ vs $2^{27}$ tokens).

**Downstream performance check** Given that fine-tuning LLMs allows us to unlock additional capabilities, such as following instructions and aligning with human behavior [19], it is crucial to assess whether the pre-training degradation also adversely affects downstream tasks. Considering that the fine-tuning dataset is typically smaller in scale, we perform fine-tuning on SQuAD [22] dataset using pre-trained checkpoints. The results, presented in Table 5, indicate that the token-crisis experienced during pre-training indeed has a detrimental impact on downstream tasks. For instance, the model trained with $2^{27}$ tokens, despite achieving a validation set score of 2.9 points in pre-training, experiences a drop of 1.9 points in F1 score for the downstream task.

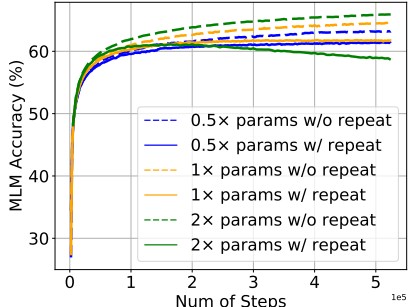

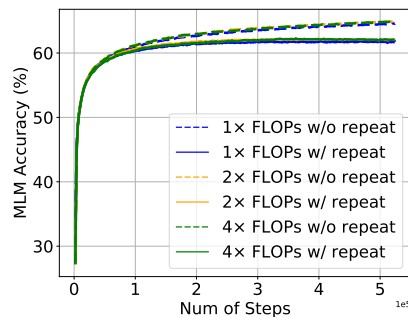

(a) We use comparable FLOPs and change the number of trainable parameters by using ParamShare with $0.5\times$ parameters, vanilla model with $1\times$ parameters and MoE with $2\times$ parameters.

(b) We fix the number of trainable parameters and change FLOPs by increasing the parameter sharing and re-using of each trainable transformer block.

Figure 5: The dash lines mean this model is trained with enough tokens for one epoch (*i.e.,* no repeated data usage). The solid lines mean that these models are trained with limited data for multiple epochs.

## 3 What Are the Key Factors Contributing to Multi-Epoch Degradation?

### 3.1 Data

**Insight (3): Dataset Size** In Figure 3, we investigated the impact of dataset size and the number of token repeats while keeping the total computation budget fixed. To further explore the importance of dataset size, we conducted another experiment where we fixed the number of token repeats and varied the dataset size. Specifically, we repeatedly used $2^{27}$ and $2^{29}$ tokens for $2^8$ times, corresponding to training the model with a batch size of 128 and 512 for $2^{18}$ steps. As depicted in Figure 4, we observed a significant overfitting phenomenon when training with $2^{27}$ tokens for $2^8$ epochs. However, when using $2^{29}$ tokens for the same number of training steps, the model did not experience degradation. Since we changed both batch size and number of tokens, to ensure a more fair comparison, we conduct another set of ablation study by fixing the batch size in Appendix G. These results indicate that employing a larger dataset can alleviate the issue of multi-epoch degradation.

**Insight (4): Dataset Quality** Taylor *et al.* [30] successfully trained a 120B model on 106B tokens for 4 epochs. They suggest that dataset quality may be a key factor to avoid overfitting, although they did not conduct experiments to validate this hypothesis. As suggested in [3, 31], we assume Wikipedia dataset [5] is our high-quality dataset. To perform a fair comparison with the C4 dataset, we sampled approximately $2^{27}$ tokens from Wikipedia and trained the model for $2^{19}$ steps. Due to the differences in pre-training data usage, we directly compare the performance on a downstream task. As presented in Table 2, the model trained on a subset of Wikipedia exhibits a similar level of degradation to the model trained on a subset of C4. This indicates that the high-quality pre-training data from Wikipedia does not alleviate the issue of multi-epoch degradation. Certainly, the low quality here is relative. For extremely low-quality data like C4 with cleaning, the data quality will probably harm the performance.

### 3.2 Model

**Insight (4) & (5): Decoupling Parameters and FLOPs** Scaling the foundation model is a crucial aspect, but it is unclear which factor plays a more significant role in multi-epoch degradation. During the scaling process, both the number of parameters and the computational cost increase. To disentangle the effects of these two factors, we introduce Mixture-of-Experts (MoE) [13] and parameter sharing (ParamShare) [4] to increase or decrease parameters with comparable computation cost. MoE allows for a substantial increase in the number of parameters while maintaining comparable FLOPs per sample. We implement a T5 MoE model based on ST-MoE [38], where an MoE layer is added every 4th transformer block, with each MoE layer consisting of 16 experts. On the other hand, ParamShare reduces the number of parameters while keeping the FLOPs fixed. Following ALBERT [12], we create a T5 ParamShare model with 6 layers of trainable parameters in both the encoder and decoder. We reuse each trainable transformer block twice, resulting in the same FLOPs as the vanilla T5 model but with only around $0.5\times$ the number of parameters.

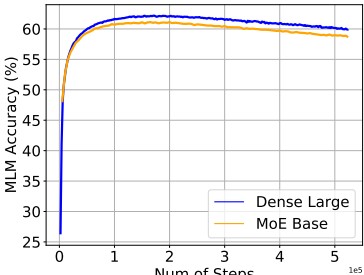
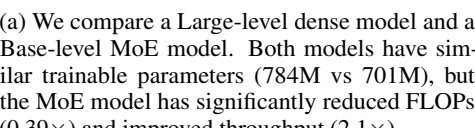
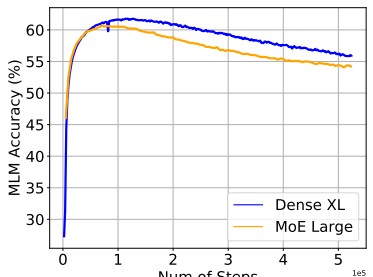

(a) We compare a Large-level dense model and a Base-level MoE model. Both models have similar trainable parameters (784M vs 701M), but the MoE model has significantly reduced FLOPs (0.39×) and improved throughput (2.1×).

(b) We compare an XL-level dense model and a Large-level MoE model. Both models have similar trainable parameters (2.8B vs 2.4B), but the MoE model has significantly reduced FLOPs (0.32×) and improved throughput (3.8×).

Figure 6: Comparing the overfitting trend of dense model and MoE model with comparable parameters but different FLOPs.

To analyze the effects of parameter variation while maintaining comparable FLOPs, we use ParamShare with $0.5\times$ parameters, vanilla model with $1\times$ parameters and MoE with $2\times$ parameters. In Figure 5a, we observe that the model with less parameters is less influenced by the usage of repeated tokens, indicating a reduced impact of multi-epoch degradation. On the other hand, the MoE model with more trainable parameters is highly prone to overfitting when trained on limited data. To further investigate the behavior of MoE models, we conduct ablation study on the number of experts in Appendix E. We found MoE models are less data-efficient and more data-hungry [11, 17, 34, 35]. While MoE can be a beneficial inductive bias when sufficient data is available, caution should be exercised when using MoE models under token-crisis or in low-resource language scenarios.

To investigate the influence of the computation budget, we fix the number of trainable parameters and vary the FLOPs by conducting experiments with three different configurations: (1) None of the 12 base-level transformer layers in both the encoder and decoder are shared. (2) We still use 12 layers of trainable parameters but reuse each layer twice, resulting in a model with $2\times$ the FLOPs compared to the baseline. (3) We further increase the computation by using 6 wider trainable layers (referred to as large-level transformer layers) and reusing each layer four times. This configuration results in a model with $4\times$ the FLOPs compared to the baseline but with comparable parameters. The results, as shown in Figure 5b, indicate that although the model with more computation achieves slightly better performance when scaling the FLOPs alone, we do not observe a clear increase in degradation.

**Insight (6): Using MoE to Predict the Behavior of Larger Dense Models** The previous findings regarding MoE models are relatively negative. However, we made an interesting observation when comparing a large-level dense model and a base-level MoE model. Despite having a comparable number of trainable parameters (784M vs. 701M), the MoE model has only around $0.39\times$ the FLOPs and $2.1\times$ the throughput. Surprisingly, these two models exhibit almost the same overfitting trend, as shown in Figure 6a. To further investigate this observation on a larger scale, we compare an XL-level dense model with a large-level MoE model. Similar to the previous finding, these models, with comparable parameters (2.4B vs. 2.8B), exhibit a similar overfitting trend, despite the large-level MoE model having only $0.32\times$ the FLOPs and $3.8\times$ the throughput.

We believe that this finding is significant. In the era of large models, training a model at the final large scale is extremely expensive. Therefore, being able to predict the behavior of larger models using smaller and more cost-effective models, as stated in GPT-4 [18], is highly desirable. With our finding, we can accurately predict the behavior of larger models using sparse MoE models with significantly lower carbon emissions. We provide an example of how this finding can be utilized in Section 5.

### 3.3 Training Objective

**Insight (7): Can Diverse Training Objectives Alleviate Multi-Epoch Degradation?** We investigate whether diverse training objectives can improve models from different aspects and alleviate the token-crisis. Specifically, we study the UL2 training objective proposed by Tay *et al.* [29], which is a mixture of widely used pre-training objectives (*e.g.,* PaLM 2), including next token prediction and masked language modeling. UL2 covers the two most important training objectives in LLM

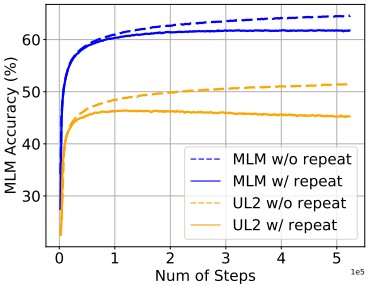

Figure 7: We compare T5 and UL2 with the same training hyper-parameters under two different settings, *i.e.,* using enough pre-training data once and repeating limited data for multiple epochs.

Table 3: We pre-train T5 and UL2 with limited data for multiple epochs. We then fine-tune these models on SQuAD to check the larger gap of UL2 pre-training performance indeed has a more negative impact on downstream tasks.

| Dataset | SQuAD | |
|---|---|---|
| Acc | EM | F1 |
| MLM w/o repeat | 82.4 | 90.0 |
| MLM w/ repeat | 79.9 (-2.5) | 88.1 (-1.9) |
| UL2 w/o repeat | 82.5 | 90.1 |
| UL2 w/ repeat | 79.6 (-2.9) | 87.6 (-2.5) |

Table 4: We conduct an ablation study on how widely used tricks, *i.e.,* dropout [26], droppath [9], label-smoothing [27], and weight decay [16] alleviate token-crisis.

| Limited Data | Dropout | DropPath | Label-Smoothing | Weight Decay | Val Acc |
|---|---|---|---|---|---|
| ✗ | ✗ | ✗ | ✗ | ✗ | 64.5 |
| ✗ | ✓ | ✗ | ✗ | ✗ | 63.6 |
| ✓ | ✗ | ✗ | ✗ | ✗ | 61.7 |
| ✓ | ✓ | ✗ | ✗ | ✗ | 62.9 |
| ✓ | ✓ | ✓ | ✗ | ✗ | 63.0 |
| ✓ | ✓ | ✗ | ✓ | ✗ | 62.6 |
| ✓ | ✓ | ✗ | ✗ | ✓ | Nan |

pre-training. Similar to the previous experiments, we use $2^{27}$ tokens for $2^8$ epochs. However, since UL2 pre-training objective is more challenging than the objective used in vanilla T5, it is unfair to directly compare the pre-training validation masked token prediction accuracy. Therefore, we report downstream results on the SQuAD dataset for reference. It is important to note that we use the same hyper-parameters as T5 for a fair comparison.

The results, summarized in Figure 7 and Table 3, compare the vanilla masked language modeling (MLM) objective in T5 with the UL2 training objective using the same training hyper-parameters. We examine both scenarios of using enough pre-training data and using limited data for multiple epochs. Although we cannot directly compare the validation accuracy of T5 and UL2, it is evident that UL2 is more prone to overfitting and exhibits a more pronounced multi-epoch degradation. For instance, when using the vanilla masked language modeling objective on the base-level model, the performance does not drop during training. However, with the UL2 objective, the validation accuracy starts to decline early on in the pre-training phase. Moreover, in the downstream evaluation (Table 3), the performance drop of UL2 is larger than that of vanilla T5. However, it is important to highlight that, although UL2 shows negative results in our token-crisis setting, it actually verifies the effectiveness of the UL2 training objective. The findings in this section indicate that UL2 can accelerate the model's learning process, which is precisely the key aspect of a well-designed LLM pre-training objective.

# 4 Can Regularization Alleviate Multi-Epoch Degradation?

**Exploring Widely Used Regularization Technologies** We explore widely used regularization technologies, including dropout [26], droppath [9], label-smoothing [27], and weight decay [16], to alleviate the multi-epoch degradation observed under token-crisis. We present the results of the ablation study in Table 4. Our findings indicate that using dropout alone is highly effective in alleviating the multi-epoch degradation. Adding additional regularizations on top of dropout does not lead to further performance improvements. In fact, introducing weight decay can even make the training process unstable. This finding is actually good news for practitioners because dropout is easy to implement with model parallelism. Although we can see a slight improvement when using label-smoothing, we do not consider it as default because label-smoothing may introduce unforeseen issues for beam search. Regarding DropPath, its effectiveness may be limited in our case due to the

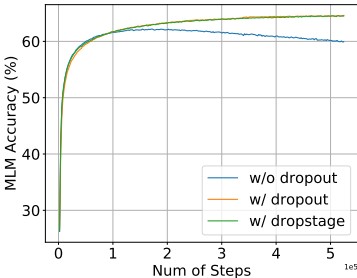

Figure 8: We train three Large-level models, including one model without dropout, one model using dropout from start, and one model using dropout after $2^{15} \approx 32K$ steps (*i.e.,* w/ dropstage).

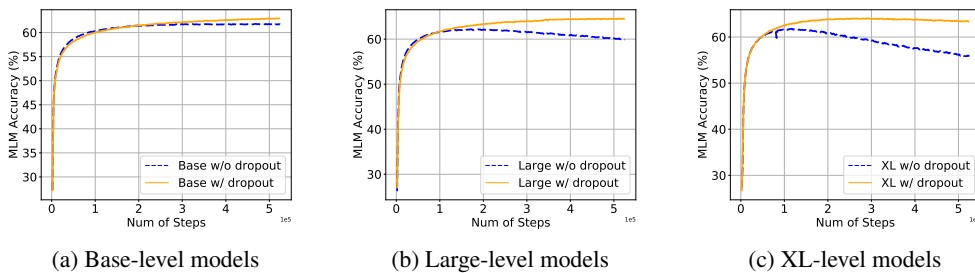

     (a) Base-level models           (b) Large-level models           (c) XL-level models

Figure 9: We set dropout as 0.0 or 0.1 when training at different scales with limited data.

use of tensor parallelism [25] during pre-training. Introducing more communication across GPU or TPU cores to support DropPath may slow down the training process and reduce hardware utilization.

**Insight (8): Dropout as an Effective yet Underutilized Regularization Technique in LLM** An interesting observation is that most existing LLMs with over 10 billion parameters, such as GPT-3, PaLM, LLaMA, Chinchilla, and Gopher, do not utilize dropout as a regularization technique. We hypothesize that the reason is that using dropout can potentially slow down the model's learning process when an ample amount of training data is available, as demonstrated in Table 4. However, it is worth noting that Galactica [30] is an exception as they incorporate dropout in their training process. This could explain why they were able to successfully train a 120 billion-parameter model without experiencing overfitting, despite emphasizing the importance of data quality as a key factor in their work. Please note We have investigated the influence of data quality in Section 3.1 and found it to be less significant than we thought.

**Insight (9): Gradual Integration of Dropout during Training** In order to ensure that the model performs well throughout the entire training process, we explore an alternative approach where dropout is introduced only at a later stage of the training process. For the early stage of training, we do not use dropout. In Figure 8, we conducted an experiment using $2^{27}$ tokens for $2^8$ epochs, where a Large-level model was pretrained for a total of $2^{19}$ steps. In the first $2^{15}$ steps, we did not employ dropout, and for the remaining $2^{19} - 2^{15}$ steps, we applied dropout with a rate of 0.1. The results show that the model with dropout introduced at a later stage performs comparably to the model with dropout from the beginning. Upon closer examination, we also observed that the model using dropout later outperforms the model with dropout from the start during the early stages of pre-training. These findings suggest that gradually integrating dropout during training can achieve comparable performance to using dropout from the beginning, while potentially offering some advantages in the initial phases of training. This approach allows for flexibility in the application of dropout, ensuring the model's performance is not compromised during the early training stage.

**Insight (10): Dropout Performance at Different Model Scales** While dropout has shown promising results at the Base-level, we aimed to investigate its effectiveness when scaling up the models. We conducted experiments using dropout with a rate of 0.1 and trained models with limited data for multiple epochs, as shown in Figure 9. The results clearly demonstrate that dropout can have a significant positive impact on performance across different scales of models. However, it is important to note that dropout is not a perfect solution, particularly when dealing with XL-scale model. Even with dropout applied, there is still a slight drop in validation accuracy at the later stages of training for the XL-level model. This suggests that while dropout can effectively mitigate overfitting and improve performance, it may face additional challenges when scaling up to larger models.

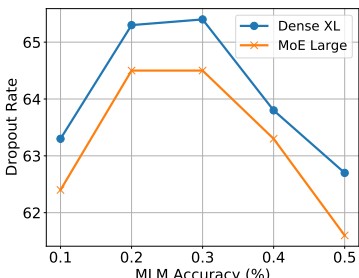

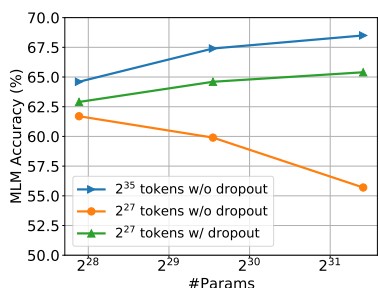

Figure 10: We first sweep dropout from 0.1 to 0.5 for Large-scale MoE and then sweep the same range for XL-scale Dense model to validate.

Figure 11: We compare the model trained with enough data and the models trained with limited data with and without dropout.

# 5 MoE Hyper-Parameter Tuning

We discovered that as we scale up, dropout may necessitate additional hyper-parameter tuning. However, conducting hyper-parameter tuning at a large scale can be extremely costly. For instance, in our specific scenario, training T5-XL five times would require approximately $37,000 USD for renting Google Cloud TPUs. Considering even larger models like PaLM and GPT-4, trained on even larger datasets, this cost becomes unmanageable. To address this issue and minimize the expense associated with hyper-parameter tuning, we leverage the insight that a Sparse MoE model can approximate the optimal hyper-parameters by predicting the behavior of a larger dense model.

**Insight (11): Determining Optimal Hyper-parameters for Dense Models through MoE Sweeping**
We first validate the aforementioned insight regarding MoE behavior prediction still holds after incorporating dropout in Appendix I. Then, to identify the optimal hyper-parameters for dense models, we employed a two-step process. First, we conducted a sweeping analysis of the dropout rate in the Large-scale MoE model across the range of {0.1, 0.2, 0.3, 0.4, 0.5}. Subsequently, we performed the same sweeping procedure for the XL-scale Dense model to validate the accuracy of the dropout rate identified by the MoE model. As illustrated in Figure 10, both the Dense XL and MoE Large models exhibit nearly identical curves. Notably, they indicate that setting the dropout rate to 0.2 or 0.3 yields the optimal performance.

This discovery holds significant practical value for practitioners. The advantage of leveraging the MoE model for hyper-parameter tuning is evident. It requires considerably fewer computational resources compared to the dense model, despite possessing comparable parameters. This translates into substantial savings in computation resources for debugging and hyper-parameter tuning. For example, in this specific set of experiments, sweeping the MoE Large model incurred an expenditure of approximately 10.6K USD on the Google Cloud Platform. Conversely, training the Dense XL model only once required 7.4K USD. Consequently, the entire development process, including sweeping, amounted to a total cost of 18K USD, which is only 0.48 times the expense of directly tuning the Dense XL model. As we scale up to larger models and conduct more experiments, the potential of MoE Hyper-Parameter Tuning to conserve computational resources and reduce carbon emissions becomes increasingly promising for future endeavors.

**Final Performance after Dropout Sweep** Having determined the appropriate dropout rate based on the MoE model, we proceed to scale up the models accordingly. Figure 11 illustrates the outcomes of this scaling process. Notably, we only used around $2^{27}$ tokens, which should be only able to train a 16M T5 model according to Chinchilla scaling law. However, we can see the model can still improve when scaling to 2.8B parameters, *i.e.,* over $1700\times$ larger than the 16M model. We believe this is a significant result for such a simple method, *i.e.,* introducing an appropriate dropout rate via MoE Hyper-Parameter Tuning.

# 6 Conclusion

In this study, we investigated the token-crisis problem and thoroughly explored various approaches to training LLMs with repeated tokens. Our investigation covered what is token-crisis, what would happen if we use repeated data under token crisis, why there is multi-epoch degradation when repeating data, and how can we alleviate this issue with off-the-shelf approaches. We also demonstrated the effectiveness of using MoE to predict the behavior of more computationally expensive dense models, offering a valuable means of accelerating LLM development more broadly.

## Acknowledge

Yang You's research group is being sponsored by NUS startup grant (Presidential Young Professorship), Singapore MOE Tier-1 grant, ByteDance grant, ARCTIC grant, SMI grant and Alibaba grant. We also thank the TPU computing resources from Google TRC (TPU Research Cloud) grant.

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

# Appendix

## A    Frequent Asked Questions

We list the potential frequent asked questions and the point-to-point answers as follows:

### A.1    Selection of Encoder-Decoder T5 Model and C4 Dataset

Firstly, the C4 dataset is a widely studied large-scale pre-training dataset that has been open-sourced. It is relatively easier to prepare and maintain compared to the dataset used in LLaMA. T5 was originally proposed and developed in conjunction with the C4 dataset. Therefore, using the T5 model with the C4 dataset aligns with established practices and facilitates comparability with prior research.

Furthermore, while decoder-only architectures have been predominant in existing Language Models (LLMs), it is still unclear whether decoder-only models consistently outperform encoder-decoder architectures. Recent research by Tay *et al.* [29] demonstrated that encoder-decoder architectures exhibit superior performance at the 20B scale. Additionally, the specific architecture employed by OpenAI's GPT-4 remains unknown to external researchers. Consequently, there is ongoing debate and uncertainty regarding the superiority of decoder-only models.

In reality, the difference between decoder-only and encoder-decoder architectures may not be as significant as initially perceived. Both architectures utilize an autoregressive decoder, and the main distinction lies in determining which tokens are fed into the encoder and which ones are fed into the decoder. Another difference arises in multi-turn dialogue systems, where encoder-decoder architectures may require recomputation of certain activations.

### A.2    Lack of Significant Improvement of UL2 in Table 3

The lack of significant improvement of UL2 over the vanilla T5 model in Table 3 can be attributed to a specific difference in our implementation. Unlike the original UL2 implementation by Tay *et al.* [29], we did not utilize dropout in our experiments. By removing dropout from the UL2 training process, it is possible that we experienced a performance drop, leading to the comparable results observed between UL2 and the vanilla T5 model in our experiments.

### A.3    Reason for Not Training a Larger Scale Model (10B+)

While training a larger-scale model like 10B+ parameters would indeed provide valuable insights, the primary reason for not doing so is the limitation of computational resources. Conducting a comprehensive set of experiments covering various aspects of the token-crisis issue requires substantial computational power, which includes significant financial costs, time requirements, and carbon emissions. Therefore, in order to balance these factors and optimize our research efforts, we decided to scale up to approximately 3B parameters to explore the token-crisis problem within the available resource constraints.

### A.4    Performance Drop Without Enough Data Even If We Are Using Dropout

Even when employing a suitable dropout rate, there is still a substantial gap between models trained with full tokens (without dropout) and those trained with a limited number of tokens through data repetition. This discrepancy arises because, for the purpose of clarity, we repeated only a small number of tokens (i.e., $2^{27}$) across multiple epochs ($2^8$ epochs). In more typical scenarios, such as repeating 1T tokens for 10 epochs, the observed gap would probably be smaller.

# B  Related Work

## B.1  Multi-Epoch LLM Pre-training

The topic of training LLMs for multiple epochs only received little attention in existing literature. Hoffmann *et al.* [8] suggested that training LLMs for multiple epochs may have detrimental effects. On the other hand, Tay *et al.* [29] trained a 120B model for 4 epochs without observing multi-epoch degradation, although they attribute their success primarily to the high-quality data used. biderman *et al.* [2] found that deduplicating pre-training data had no clear benefit on language modeling performance. Additionally, Hernandez *et al.* [7] discovered that repeating a small fraction of data during LLM pre-training can significantly harm model performance.

In contrast to these works, our study focuses specifically on the token-crisis problem and investigates the consequences of further scaling LLMs by repeating a fixed amount of data multiple times. To the best of our knowledge, ours is the first paper to explore the token-crisis and train LLMs for multiple epochs. Many of the insights we present, such as using Mixture-of-Experts (MoE) models to predict the behavior of more computationally expensive dense models, are novel and valuable contributions to our research community.

## B.2  Pre-training with Synthetic Data

Existing research has examined the concept of pre-training LLMs with synthetic data as a means of mitigating data scarcity. For instance, Ri *et al.* [23] designed artificial languages with structural properties that mimic natural language, while Wu *et al.* [33] successfully pre-trained LLMs through simpler synthetic tasks. Other works, such as TAPEX [15], address the challenge of data scarcity in specific domains.

In contrast, our paper focuses on training LLMs in the context of the token-crisis problem with multiple epochs, which is distinct from the perspective of using synthetic data. However, we believe this line of research is relevant because it has the potential to alleviate the token-crisis in LLMs, thereby addressing the associated challenges and limitations.

# C  Limitations

**Scalability to State-of-the-Art Models:** Although we conducted experiments on models with 3B parameters, we acknowledge that we did not explore the performance of our approach on state-of-the-art scale models, such as the 175B-parameter GPT-3. The primary reason for this limitation is the lack of computational resources required for running experiments on very large models multiple times. Given the token-crisis scenario and the need for extensive experimentation, conducting experiments on larger models remains prohibitively expensive.

**Dataset Quality Assumptions:** Our Insight (4) relies on using C4 as low-quality data and Wikipedia as high-quality data. While we acknowledge that the data quality of C4 is acceptable, we adopted this setting because many existing LLMs have utilized Wikipedia for pre-training for at least one epoch, and this choice is supported by claims of better data quality. It is worth noting that other datasets with different characteristics may yield different results when applied to our approach.

**Limitations of Insight (6) in Handling FLOPs Disparities:** Our Insight (6) may encounter challenges when dealing with significant gaps in FLOPs between models. For example, we found that the Base-level MoE model with 64 experts in every MoE layer cannot perfectly predict the behavior of an XL-level dense model, although it still exhibits a higher susceptibility to overfitting compared to the Base-level MoE model with 16 experts. This observation aligns with the notion that very sparse models may have under-trained parameters, leading to imperfect predictions. Further investigation is necessary to address this limitation and enhance the accuracy of Insight (6) when confronted with substantial FLOPs disparities.

# D  Ablation Study on Batch Size of Larger Dataset Size

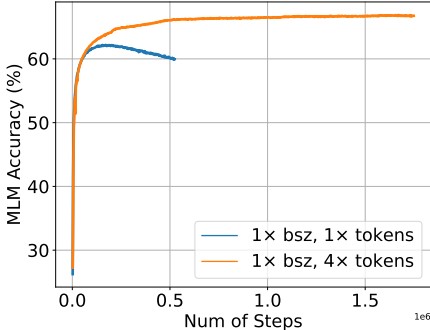

Figure 12: We fix the batch size and number of repeats, and then train the model with 4 times more steps to go through 4 times more tokens.

We can see when using smaller batch size than what we did in Figure 4, model still has little multi-epoch degradation. This further verifies that larger dataset can alleviate the multi-epoch degradation.

# E  Ablation Study on Number of Experts for MoE

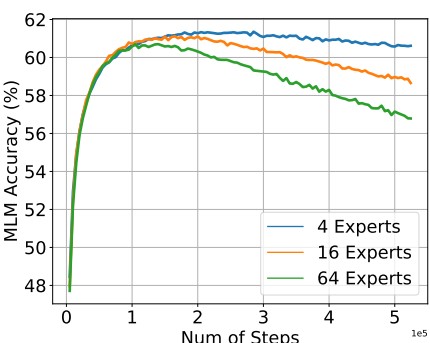

Figure 13: we use relatively limited tokens to train three T5-MoE models with 4, 16, 64 experts at each MoE layer. We can observe MoE with more experts, which have more trainable parameters have a more serious multi-epoch degradation.

To further investigate the behavior of MoE models, we train three T5-MoE models with 4, 16, and 64 experts using a relatively limited number of tokens[6]. In Figure 13, we observe that as the number of trainable parameters increases with more experts, the MoE models experience more severe multi-epoch degradation. These findings indicate that the additional parameters in MoE models are indeed leading to faster memorization of the training data.

# F  Encoder-Decoder vs Decoder-Only

We can see encoder-decoder is clearly better than decoder-only but the MoE based decoder-only model having comparable trainable parameters with encoder-decoder performs almost the same as

---

[6]Following the convention in [38], the number of experts represents the number of expert FFNs in each MoE layer. In our case, we use one MoE layer every 4 layers in both the encoder and decoder, resulting in a total of 6 MoE layers in the T5 MoE Base model. For more details about the MoE hyperparameters, please refer to Appendix J.

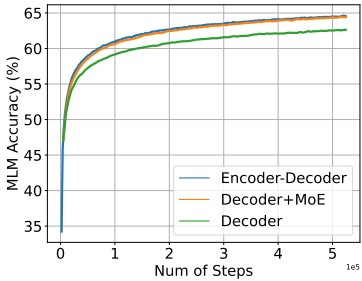

Figure 14: We train three models (*i.e.,* encoder-decoder, decoder-only, MoE-based decoder-only) with the same data (C4) and training objective (Span-Corruption).

encoder-decoder model. Therefore, as suggested by UL2 paper, the different behaviours of encoder-decoder and decoder-only are more from the training objective instead of model architecture. That is the reason why we explore UL2 training objective in our Section 3.3.

# G  Ablation Study on Batch Size of Larger Dataset Size

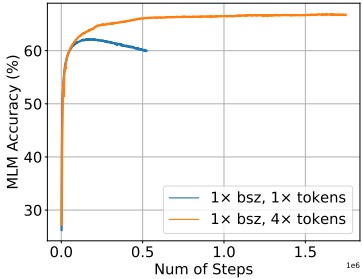

Figure 15: We fix the batch size and number of repeats, and then train the model with 4 times more steps to go through 4 times more tokens.

We can see when using smaller batch size than what we did in Figure 15, model still has little multi-epoch degradation. This further verifies that larger dataset can alleviate the multi-epoch degradation.

# H  Downsteam Evaluation

Table 5: The updated fine-tuning results. We include the standard deviation of 5 runs.

| Model | BoolQ | RTE | SQuAD | |
|---|---|---|---|---|
| | Acc | Acc | EM | F1 |
| C4-SpanCorr-Repeat1 | 77.0±0.7 | 68.9±0.8 | 82.3±0.9 | 90.0±0.9 |
| C4-SpanCorr-Repeat2$^8$ | 73.8±1.0 | 66.4±0.3 | 80.0±0.6 | 88.1±0.5 |
| C4-UL2-Repeat1 | 78.8±0.9 | 72.2±1.2 | 82.3±0.8 | 90.2±0.7 |
| C4-UL2-Repeat2$^8$ | 74.5±0.6 | 68.4±0.7 | 79.3±1.1 | 87.3±1.2 |
| C4-Wiki-Repeat1 | 74.2±0.7 | 69.3±1.1 | 82.6±0.8 | 90.1±0.6 |
| C4-Wiki-Repeat2$^8$ | 71.4±0.4 | 64.9±0.5 | 79.6±1.0 | 87.9±1.3 |

# I  Verification of MoE Behavior Prediction with Dropout

To validate the aforementioned insight regarding MoE behavior prediction, we conducted an examination incorporating dropout, as depicted in Figure 16. Remarkably, our findings reveal that the MoE model maintains a remarkably similar training behavior to that of the dense model, even when utilizing additional computation resources and possessing comparable parameters. This observation

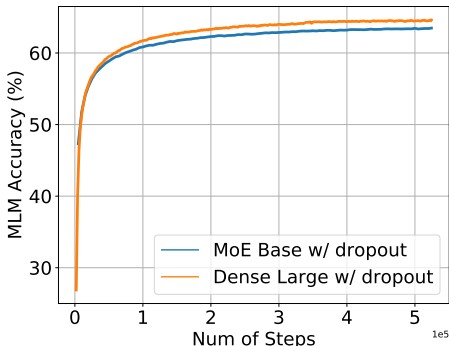

Figure 16: We add dropout to train two models with comparable trainable parameters, *i.e.,* Large-level Dense model and Base-level MoE.

further reinforces the effectiveness of the MoE model in approximating the behavior of larger, denser models.

## J   Default Hyper-parameters

Table 6: Default pre-training hyper-parameters.

| Name | Value |
| --- | --- |
| Learning Rate | 0.01 |
| Learning Rate Decay | Square Root Decay (0.8) |
| Optimizer | Adafactor |
| Batch Size | 128 |
| Training Steps | 524288 ($2^{19}$) |
| Dropout | 0.0 |
| DropPath | 0.0 |
| Label Smoothing | 0.0 |
| Weight Decay | 0.0 |

Table 7: Default MoE hyper-parameters.

| Name | Value |
| --- | --- |
| Num of Experts | 16 |
| MoE Layer Layout | Every fourth |
| Train Capacity Factor | 1.25 |
| Test Capacity Factor | 2.0 |
| Num Selected Experts | 2 |
| Router Weight | 1e-2 |
| Z-loss Weight | 1e-4 |

We follow T5 1.1 implementation in T5x and ST-MoE implementation in Flaxformer. The default hyper-parameters are shown in Table 6 and Table 7. The pre-training parameters in Table 6 are used in both dense and MoE model training.

We conduct experiments on Google Cloud TPU. For Base-level and Large-level models, we use 32 TPU v3 cores, and 128 TPU v3 cores are employed to train XL-level models.

# K   Model Configurations

Table 8: Default MoE hyper-parameters.

| Scale | # Enc Layers | # Dec Layers | # Heads | Hidden Dim | MLP Dim | #Params |
|---|---|---|---|---|---|---|
| Small | 8 | 8 | 6 | 512 | 1024 | 78M |
| Mid | 10 | 10 | 10 | 640 | 1280 | 140M |
| Base | 12 | 12 | 12 | 768 | 2048 | 247M |
| Base Plus | 16 | 16 | 14 | 896 | 2560 | 432M |
| Large | 24 | 24 | 16 | 1024 | 2816 | 783M |
| Large Plus | 24 | 24 | 22 | 1408 | 4096 | 1.4B |
| XL | 24 | 24 | 32 | 2048 | 5120 | 2.8B |

For Base, Large, XL scale, we follow the default setting in T5x. To obtain a more smooth scaling curve, we add more configurations like Small, Mid, Base Plus and Large Plus. Please note the XL scale was not used to draw the scaling curve in Figure 2 because of the expensive training cost.

# L   More Results

In this section, we share more other approaches we explored.

## L.1   Training with Larger or Smaller Batch

Empirically, larger batch size is easier to overfit, and smaller batch size can usually reduce overfitting. We found this widely-used commonsense still holds in LLM token-crisis. However, we think this trick is not that useful because training LLM with smaller batch size is inefficient.

## L.2   Training with Longer or Shorter Sequence

We train the same amount of tokens with longer or shorter sequences with the same total token usage. That is, for longer sequence, we train for fewer steps and more steps are used to train with shorter sequences. We found using longer sequences can achieve better performance.

## L.3   Randomly Activating One Embedding Layer from Multiple Embedding Layers

We suggested the over-fitting can be alleviated when adding more randomness into input embeddings. We therefore init multiple embedding layers and randomly activate one of these embedding layers for each input sequence. However, we did not observe this design can alleviate multi-epoch degradation even if we regularize different embedding layers have different embeddings.

## L.4   Adding Dropout Before and After Linear Layers

Considering dropout is very effective, we tried to add dropout before and after each linear layer. Adding dropout after one layer means a column-wise weight masking. Adding dropout before and after one layer denotes element-wise weight masking. We found such regularization is too strong and makes LLM achieve inferior performance even if we set a small dropout rate on the XL-level model.

