# OpenReview forum: "To Repeat or Not To Repeat: Insights from Scaling LLM under Token-Crisis"
_NeurIPS.cc/2023/Conference — NeurIPS 2023 poster_

### Official Review · Reviewer_2hBw · 2023-07-01

**Soundness:** 3 good
**Presentation:** 3 good
**Contribution:** 3 good
**Rating:** 7
**Confidence:** 4

**Summary:**

This paper conducts an empirical study on the scaling of transformer models under limited training data, which they call a *token crisis*. They show that under token crisis, training T5 models for multiple epochs results in the degradation of pre-training and downstream task performance. They also show that the dataset quality is unimportant for this multi-epoch degradation. They reveal that using dropout can alleviate the multi-epoch degradation. They also observe that the behavior of Mixture-of-Expert (MoE) models can be used to predict the training behavior of dense models, and they use this observation to search for the best dropout rate for the dense model using MoE models.

**Strengths:**

1. This paper is well-written, and the empirical takeaways are clear and useful
2. The *token crisis* is important and is expected to be more severe. It is good that someone studies this problem.

**Weaknesses:**

1. This paper only uses a single task as the downstream task, so it is unclear if the result will hold for other datasets.
2. This paper only studies encoder-decoder models like T5, so it is unclear if the results will hold for other models. Specifically, decoder-only models (like GPT3, Chinchilla, and LLaMA) are more widely used and are the predominant architecture of current LLMs. Still, I think this paper has merits since for focusing on Enc-Dec models, and such a study is still valuable.
3. Some conclusions are too assertive and are not fully supported by the experiment results.
    - Section 3.1 Insight (3) attributes the performance difference between the two models to dataset size. However, the batch size is also different. I wonder if the key difference is the dataset size or the batch size.
    - Section 3.3: UL2 degrades more. There should be some statistical significance comparison or variance for the downstream performance based on multiple runs shown in Table 3.

**Questions:**

- Q1. Line 249: balance between regularization and model performance. Doesn't regularization mean less overfitting, which leads to better model performance? What does the balance mean here? Should it be the balance between regularization and training efficiency?
- Q2. Does Chinchilla scaling law really hold for Enc-Dec models like T5? Chinchilla law is based on Decoder only models. More extensive experiments (more than the experiments shown in Insight (1)) are needed to verify such a claim.
- Q3. The sentence on Line 111 (`When a larger model outperforms a smaller model, it indicates that the smaller model has received sufficient tokens`) is unclear to me. I don't see why the prior situation indicates the latter conclusion.
- Q4. Line 153: Why is C4 with cleaning still extremely low-quality?

Presentation
===
Line 244: "We" should not be capitalized.


Other suggestion
===
**The following comment does not influence my rating of the paper, and I know that the authors are not required to compare their work with the following concurrent work**. I am just listing it since I believe that when the paper under review is published, readers will be wondering the following questions.
Another very recent work also studies token crisis, "Scaling Data-Constrained Language Models [1]". It will be beneficial to discuss the similarity and differences between that paper and the current paper under review. For example, why does the Chinchilla law need to be adjusted in [1] while the current paper claims that the Chinchilla law still holds?

**Limitations:**

The limitations are well-addressed.

---

> ### Author Rebuttal · Authors · 2023-08-09
>
> > Q1: This paper only uses a single task as the downstream task, so it is unclear if the result will hold for other datasets.
>
> Thank you for your insightful suggestion. We conducted more downstream evaluation on BoolQ and RTE datasets, which are two widely used datasets in SuperGLUE benchmark. As shown in the rebuttal PDF, we observe a similar trend in both SuperGLUE and SQuAD results.
>
> > Q2: This paper only studies encoder-decoder models like T5, so it is unclear if the results will hold for other models. Still, I think this paper has merits because for focusing on Enc-Dec models, and such a study is still valuable.
>
> Thank you so much for your positive feedback. Please see the response of SEVe Q1 and Appendix A.1.
>
> > Q3: Section 3.1 Insight (3) attributes the performance difference between the two models to dataset size. However, the batch size is also different. I wonder if the key difference is the dataset size or the batch size.
>
> This is a great question! We conduct another set of experiments in rebuttal pdf file by fixing the batch size and let the model go through the 4 times larger dataset. We can see that model trained by fixed batch size has a similar trend to the model trained with a 4 times larger batch size.
>
> > Q4: Section 3.3: UL2 degrades more. There should be some statistical significance comparison or variance for the downstream performance based on multiple runs shown in Table 3.
>
> We added the standard deviation in the PDF submission.
>
> > Q5: Line 249: balance between regularization and model performance. What does the balance mean here?
>
> Dropout serves as a regularization technique to reduce overfitting in LLM training to improve model performance. However, it can slow down early learning. The "balance" refers to optimizing both training efficiency and model performance through the whole training process. To achieve this balance, we experimented with applying dropout mainly in later epochs to prevent overfitting, while avoiding its use in the initial training stages. We've clarified this point in the latest version of our paper and will update it accordingly.
>
> > Q6: Does Chinchilla scaling law really hold for Enc-Dec models like T5? Chinchilla law is based on Decoder only models. More extensive experiments (more than the experiments shown in Insight (1)) are needed to verify such a claim.
>
> First, we argue again that Decoder-only is not that different from Enc-Dec models. In addition, we check the Chinchilla scaling law in our paper because we want to see whether encoder-decoder architecture is also similarly data-hungry. Through our experiments depicted in Figure 2, we indeed observed that larger models outperform smaller ones given a fixed computation budget, with the requirement for an increased dataset size. This observation reinforces the necessity of investigating multi-epoch training. To ensure clarity and precision in our assertion, we have revised our claim as follows: "Encoder-Decoder models trained on the C4 dataset exhibit comparable data-hungry behavior as described in the Chinchilla scaling law.". This would be reflected in our forthcoming version of the paper.
>
> > Q7: The sentence on Line 111 (When a larger model outperforms a smaller model, it indicates that the smaller model has received sufficient tokens) is unclear to me. I don't see why the prior situation indicates the latter conclusion.
>
> Larger model is more data-hungry so the larger model is superior only after learning from enough tokens. > Q7: The sentence on Line 111 (When a larger model outperforms a smaller model, it indicates that the smaller model has received sufficient tokens) is unclear to me. I don't see why the prior situation indicates the latter conclusion. Larger model is more data-hungry, which means larger model is superior only after learning from enough tokens. Assume we get more computation budget and infinite training tokens, the question here is whether we should train a larger model or just train a smaller model for longer? Since smaller model is cheaper during inference, we only want a larger model when the larger model is better than the smaller one given the same training cost. Since smaller model has relatively limited capacity to consume more data, it would be outperformed when we have enough resource to train larger model for long enough. That is the case that the smaller model has received enough data and we should train a larger model instead.
>
> > Q8: Line 153: Why is C4 with cleaning still extremely low-quality?
>
> Sorry for the confusing statements. As we discussed in Appendix C, the terms "high" and "low" quality are relative, and our classification of C4 as low-quality is made in comparison to Wikipedia. To prevent any misunderstanding, we will incorporate a footnote in our upcoming version.
>
> > Q9: Discuss the concurrent token crisis paper, "Scaling Data-Constrained Language Models [1]".
>
> Thank you so much for pointing this out! We are very happy when seeing this concurrent work focusing on token-crisis and glad to discuss the difference.
>
> While both our paper and the referenced work address token-crisis concerns, they diverge in focus. The other work primarily delves into the specific scaling law associated with repeated data usage—identifying the optimal repetition times based on model and dataset sizes. In contrast, our primary objective is to scrutinize the multi-epoch scaling behavior. Notably, we explore influential factors like UL2 and MoE techniques, which exhibit heightened data dependency, and we discover the surprising effectiveness of dropout in mitigating multi-epoch degradation.
>
> These papers offer orthogonal insights into the token crisis problem. The other work contends that the Chinchilla scaling law necessitates adjustment due to the use of repeated data—a distinct context from the original Chinchilla paper. In contrast, our experimentation with T5 does not involve data reuse, maintaining a setting more akin to the Chinchilla study.

---

> > ### Comment · Reviewer_2hBw · 2023-08-15
> > **Re: Rebuttal**
> >
> > Thank you for your detailed responses. I carefully read the responses, and most of my questions are answered. I appreciated the supplementary experiments, and I encourage the authors to add them to the final version should this paper be accepted.
> >
> > Still, I do not get the responses to question 7. Reading Section 2 Insight (1) again makes me unsure of what this paper is doing. Here, the paper tries to verify if Chinchilla scaling law applies to T5 when training on C4. However, the experiment here compares T5 of different sizes (6 different models) when training with different numbers of tokens (and I am not sure how the optimal number of tokens is calculated) and training at various computation budgets (according to Line 110). How can we fit the Chinchilla scaling law using the above experiment?
> > In the Chinchilla paper, they either fix the model size and vary the number of training tokens or fix the computation budget (FLOPs) and vary the model size. Last, they use all those models to fit the Chinchilla law. However, it seems to me that all those factors vary in this paper, so it is unclear to me how the scaling law is verified here.
> >
> > Can the authors elaborate more on how they conduct the experiments, how they calculate the optimal number of tokens in Figure 2, and how this supports the claim that "Chinchilla law holds on T5"?

---

> > > ### Author Response · Authors · 2023-08-15
> > > **Re: Re: Rebuttal**
> > >
> > > We really appreciate your careful review!
> > >
> > > And Sorry for the confusing statements. We devote to checking "Encoder-Decoder models trained on the C4 dataset exhibit comparable data-hungry behavior as described in the Chinchilla scaling law." Chinchilla's paper figured out the specific scaling law in their paper, but what we want to do is checking a similar trend exists in our setting, and this is to make sure the following experiments are meaningful.
> > >
> > > Therefore, we used a simplified (cheaper) way:
> > >
> > > Since T5 models are trained by inverse square root LR schedule instead of the cosine LR schedule used in Chinchilla, to get different checkpoints with varied costs and fixed model size, we do not need to run the same model size many different times by setting the "max_cosine_schedule_steps" before training. Therefore, within a single training run, we can use different checkpoints during training as the models with the fixed model size and varied training tokens. For instance, a T5-Base trained with 5K steps is exactly the same as the T5-Base trained with 10K steps' checkpoint at 5K steps. And similarly, if we consider the case of "fix the computation budget (FLOPs) and vary the model size", we can use a smaller model checkpoint trained with more steps and a larger model checkpoint trained with fewer steps. These two checkpoints consumed the same FLOPs but have different model sizes. We will add this explanation to our submission.
> > >
> > > Hope this explanation solved your concern well. Thank you so much for your careful review and comments again. Your comments improved this draft a lot!

---

> > > > ### Comment · Reviewer_2hBw · 2023-08-15
> > > > **Re: Re: Re: Rebuttal**
> > > >
> > > > I highly appreciate the authors' clear explanations. They almost clarify all my questions.
> > > >
> > > > The last question is, it seems that this paper tries to say in Section 2 Insight (1) that "the larger the model requires more tokens to reach the optimal loss" and use this to support the claim that "Chinchilla scaling law applies to T5" (in the caption of Figure 2). Can you tell me how I can arrive at a similar conclusion using the Chinchilla law in the Chinchilla paper?
> > > >
> > > > And a less minor question is the caption of Table 7 is odd. I assume there is no MoE here?

---

> > > > > ### Author Response · Authors · 2023-08-15
> > > > > **Re: Re: Re: Re: Rebuttal**
> > > > >
> > > > > Thank you for the questions!
> > > > >
> > > > > Let's use this Chinchilla preprint: https://arxiv.org/pdf/2203.15556.pdf
> > > > >
> > > > > In Chinchilla paper's Fig 2 (left), we can see larger models require more FLOPs to surpass smaller models. And certainly, for larger models, more training FLOPs do not always mean more training tokens, so we should further take a look at Table 2 (maybe also Figure A3 and Table A3). These results show that the more training FLOPs used by larger models require more training steps (or tokens). In other words, if we do not increase the tokens (or training steps), larger models cannot achieve their optimal training FLOPs. This is "the larger the model requires more tokens to reach the optimal loss" insight in Chinchilla paper.
> > > > >
> > > > > The next question is how many extra training tokens we need. In our paper, we show a linear relationship in Table 2, which is very similar to Chinchilla's Figure A3. And this is used to support our claim "Encoder-Decoder models trained on the C4 dataset exhibit comparable data-hungry behavior as described in the Chinchilla scaling law."
> > > > >
> > > > > For Table 7's typo, yes, there is no MoE. Thanks for pointing it out! We will fix that. And thanks a million for spending time on our paper and even reaching the last page of our Appendix!

---

> > > > > > ### Comment · Reviewer_2hBw · 2023-08-15
> > > > > > **Re: Re: Re: Re: Re: Rebuttal**
> > > > > >
> > > > > > Thank you. It is excellent to interact with the authors, and I have learned a lot from the discussions. Your explanation clarifies my questions.
> > > > > >
> > > > > > I will not adjust my initial ratings since they are already quite strong.
> > > > > >
> > > > > > Last, I would like to point out that a paper does not need to conduct every possible experiment to get acceptance. Focusing on a good topic and conducting a detailed analysis deserve acceptance. **What I want to say is that I don't think the paper should be rejected because it studies the encoder-decoder model instead of the widely used encoder model.** Who knows if, one day, someone will discover that encoder-decoder models are more useful than decoder-only models? For example, I know that there is an ICML paper that says the encoder-decoder model with multitask tuning is better than decoder-only models with multitask fine-tuning. We'll never know. The experiments in this paper are valuable, and the empirical takeaways are clear and useful. And that's what I want to see in a 9-page paper.

---

### Official Review · Reviewer_vyo8 · 2023-07-03

**Soundness:** 2 fair
**Presentation:** 3 good
**Contribution:** 2 fair
**Rating:** 3
**Confidence:** 4

**Summary:**

The paper presents an empirical study on the effect of training on multiple epochs in the data limited regime. They show that Chinchilla's scaling laws holds for T5 style models. The authors show that repeated tokens result in degradation of accuracy. The also study some of the factors contributing to this degradation. Towards the end, they show that MoEs can used a proxy to tune the hyperparameters of the larger models.

**Strengths:**

- a detailed empirical study
- the results are carefully studied

**Weaknesses:**

- Some of the conclusions are trivial. For example, larger models are more susceptible to overfitting.
- MoE is presented as a way to tune hyperparameters of larger models but it is not clear to what extent. For example, shoiuld MoE always be iso-parameter with the base dense model?

**Questions:**

- Have the authors considered the effect of learning rate in overfitting?

---

> ### Author Rebuttal · Authors · 2023-08-09
>
> > Q1: Some of the conclusions are trivial. For example, larger models are more susceptible to overfitting.
>
> Thanks for the suggestion. We argue that, although some of our conclusions are intuitively trivial, these conclusions play an important role in studying the token-crisis problem. For instance, to ensure that our investigation on relatively fewer tokens and moderate-size models can adapt better to train larger models with more data for fewer epochs, verified the insight that larger models are more susceptible to overfitting. Therefore, these conclusions not only provide insights to readers but also pave the way to more in-depth and insightful conclusions.
>
> > Q2: MoE is presented as a way to tune hyperparameters of larger models but it is not clear to what extent. For example, should MoE always be iso-parameter with the base dense model?
>
> We cannot say MoE is always iso-parameter with the base dense model before studying all hyper-parameters. And as we know, it is prohibitively expensive and almost impossible to ablate all hyper-parameters even if for a moderate-size model in LLM training. However, in this paper, we studied MoE is iso-parameter with more computation-heavy dense model at three different scales (Base, Large, XL) and under six different dropout ratios (0, 0.1, 0.2, 0.3, 0.4, 0.5). This indicates a strong trend towards iso-parameter behavior with MoE, so this suggests MoE is a promising approach for efficient hyperparameter tuning.
>
>
> > Q3: Have the authors considered the effect of learning rate in overfitting?
>
> Thank you for your insightful query. We fully agree that, as a general principle, training with an excessively small learning rate over an extended duration has the potential to exacerbate the overfitting issue. Therefore, we delved into the impact of learning rate on overfitting through an ablation study involving both smaller and larger learning rates. Our investigation revealed that while the learning rate does influence overall model performance, it only exerts a limited effect on altering the overfitting trend in our experiments.

---

### Official Review · Reviewer_kBHt · 2023-07-05

**Soundness:** 3 good
**Presentation:** 3 good
**Contribution:** 3 good
**Rating:** 6
**Confidence:** 3

**Summary:**

The authors propose a concept called token-crisis, which means the growth rate of high-quality text data available is much slower than the growth rate of data required by LLMs. This paper is the first empirical study of the repeating pre-training data for the token-crisis problem. Some major findings are: larger models are more prone to overfitting and would affect downstream tasks; using dropout is an effective way to alleviate the multi-epoch degradation, and setting the dropout rate to 0.2 or 0.3 yields the optimal performance.

**Strengths:**

1. The experiments are very comprehensive, and most of the conclusions are well-supported.

2. Many finds in the paper provide valuable insights to train better open-source LLMs in academia with limited resources.

3. The dropout mitigate token-crisis findings seem to be pretty useful.

**Weaknesses:**

1. Why there are no experiments testing for a smaller number of data repetitions? The smallest number of repetitions in the paper is 2^8, which is very large.

2. The paper seems to not answer the question in the title: to repeat or not to repeat. If we have limited data, should we repeat or not in pre-training? There seems to be no experiment showing how many repetitions are optimal with the same set of data.

3. The dataset-quality-does-not-matter-much conclusion seems to be a bit not-well-supported. If the Wiki data is of higher quality than C4 data, why the downstream performance of Wiki pre-trained without repeating model does not outperform the C4 pre-trained model? Considering there is a recent paper [1] that claims that true high-quality data helps a lot in code generation, I'm a bit unsure about this conclusion.

4. Does model architecture matters? All experiments use T5, an encoder-decoder model. Do all conclusions hold when using a decoder-only model?

[1] Gunasekar, S., Zhang, Y., Aneja, J., Mendes, C.C.T., Del Giorno, A., Gopi, S., Javaheripi, M., Kauffmann, P., de Rosa, G., Saarikivi, O. and Salim, A., 2023. Textbooks Are All You Need.

**Questions:**

1. See Weaknesses.

2. Since most of the experiments are done on the C4 dataset, which is known to be low quality, I'm wondering if the conclusions would still hold when using high-quality data. I think there currently is a trend to utilize higher quality data (whether generated from GP4 or hand-crafted) to fine-tune or pre-train smaller LLMs, which seems to obtain very good results.

**Limitations:**

The limitation section is pretty honest and actually echoes some of my concerns.

---

> ### Author Rebuttal · Authors · 2023-08-09
>
> > Q1: Why there are no experiments testing for a smaller number of data repetitions? The smallest number of repetitions in the paper is 2^8, which is very large.
>
> Thank you for your question. In our plots, we actually have the experiments testing for a small number of data repetitions. For instance, in Figure 3, we trained Base, Large, and XL models with a larger dataset (four times larger) and a reduced number of epochs (e.g., 2^6). Furthermore, in several of our figures (e.g., Fig 4, 5, 6, 7), the plotted validation performance spans the entire training process, encompassing datapoints from the initial stages, which actually involve only a few epochs of data.
>
>
> > Q2: The paper seems to not answer the question in the title: to repeat or not to repeat. If we have limited data, should we repeat or not in pre-training? There seems to be no experiment showing how many repetitions are optimal with the same set of data.
>
> Thanks for the great question again! We actually make our conclusion through the investigation but did not clearly highlight this in the conclusion section. To address this, we revised the conclusion section of our paper to provide a more explicit summary of our findings. Specifically, if we do not conduct any tricks, according to our experiments in Figure 4,5,6,7, it is okay to repeat for a few epochs. A concurrent work [6] also figured out the specific scaling law of training with repeated data. In this paper, we think studying the factors of scaling LLM under token crisis is also highly important because we find the scaling law (or the optimal repetitions) is very sensitive to many factors like dataset size, model size, training objective, and regularizations. Based on our findings, in Figure 9, we can see simply setting dropout as 0.1 can greatly change the mult-epoch degradation so that the scaling law would also be very different after adding dropout. Therefore, if we add an appropriate dropout for multiple epoch LLM training, it would be okay to repeat.
>
> [6] Muennighoff, Niklas, et al. "Scaling Data-Constrained Language Models." arXiv preprint arXiv:2305.16264 (2023).
>
>
>
> > Q3: The dataset-quality-does-not-matter-much conclusion seems to be a bit not-well-supported. If the Wiki data is of higher quality than C4 data, why the downstream performance of Wiki pre-trained without repeating model does not outperform the C4 pre-trained model? Considering there is a recent paper (Textbook is all you need) that claims that true high-quality data helps a lot in code generation, I'm a bit unsure about this conclusion.
>
> This is a great question. We also discussed the dataset quality assumption in Appendix C. Since Wikipedia dataset is actually smaller than C4, even if we use the full wikipedia dataset, the model is actually trained for a few epochs when training with 500K steps. However, the C4 is large enough so that there is no repeat at all. Therefore, it is reasonable that wikipedia is slightly worse than C4 when training with enough data.
>
> For the Textbook is all you need paper, the data quality is extremely high, which is also very similar to the downstream tasks like HumanEval (See Section 6 of the Textbook paper). So using such high-quailty instruction following data without web-scale pre-training data to achieve good performance on some benchmarks is possible. However, in this paper, we are discussing the quality of web-scale pre-training data. Note the high quality here is relative. Compared with C4, wikipeida is good. But if we compare wikipedia with Phi-1 or Vicuna’s ShareGPT instruction following data, wikipedia is relatively not that ideal.
>
>
> > Q4: Does model architecture matter? All experiments use T5, an encoder-decoder model. Do all conclusions hold when using a decoder-only model?
>
> Please see the response for Reviewer SEVe Q1 and our Appendix A.1.
>
> > Q5: Since most of the experiments are done on the C4 dataset, which is known to be low-quality, I'm wondering if the conclusions would still hold when using high-quality data. I think there currently is a trend to utilize higher quality data (whether generated from GP4 or hand-crafted) to fine-tune or pre-train smaller LLMs, which seems to obtain very good results.
>
> We argue that, firstly, C4’s quality is relatively low when compared with wikipedia but it doesn’t mean C4 is a terrible dataset. Note that LLaMA also used C4 as part of its pretraining data. As we mentioned above, The textbook data works on HumanEval because its data is similar to HumanEval benchmark to some extent. If we want to get a model which is good at everything, it is still better to pretrain the model with web-scale dataset. The instruction finetuning is out of the scope of this paper. Due to the popularity of ChatGPT, users are writing instructions every day. We can even mine instructions in the daily conversation or movie transcript and then generate a huge amount of instructions. So the number of instructions would grow fast recently and it has not approached the upper bound. In other words, it is too early to discuss the token-cirsis of instruction following data. We think it is better to wait for what would happen in the instruction following data and see whether we will really run out of instructions before studying “instruction-crisis”.

---

> > ### Comment · Reviewer_kBHt · 2023-08-19
> >
> > Thank you for the rebuttal. I feel overall positive about the paper. I raised my score to 6. Below are some additional comments:
> >
> > 1. Table 2 seems to suggest that C4 and Wikipedia have the same number of tokens (2^27), which is not consistent with the author's rebuttal, which suggests that the two datasets are not of the same sizes. If that's the case, why not use a subset of C4 that has the same size as Wikipedia data to perform the data quality experiment? It doesn't seem to be able to reach any conclusion on data quality when the two datasets are different in size.
> >
> > 2. About the model architecture comment, I didn't mean that encoder-decoder models are not as good as decoder-only models. I'm just curious whether the model architecture plays a role in the multi-epoch training degradation. The authors could include some discussion on this in the paper.

---

> > > ### Author Response · Authors · 2023-08-20
> > > **Re: Official Comment by Reviewer kBHt**
> > >
> > > We appreciate your reconsideration of the evaluation score!
> > >
> > > Sorry for the confusing statements in Table 2. Since Wikipedia has fewer tokens in total, even if we use the full dataset, the num of epochs should be larger than 1 from the start, which means the dataset size gap between full dataset training and subset training is smaller. For instance, we are actually comparing:
> > >
> > > C4($2^{35}$tokens $\times$ $2^{0}$ epoch) vs C4($2^{27}$tokens $\times$ $2^{8}$ epoch)
> > >
> > > Wiki($2^{34}$tokens $\times$ $2^{1}$ epoch) vs Wiki($2^{27}$tokens $\times$ $2^{8}$ epoch)
> > >
> > > The noteworthy observation from Table 2 is that despite Wikipedia having a smaller dataset size gap, it experiences a comparatively larger performance degradation. This insight strengthens our argument. Rather than rendering our conclusion incorrect, this observation further reinforces its validity.
> > >
> > > Sorry again for the confusing statements in the paper. We will add this explanation and the discussion about encoder-decoder vs decoder only in our paper. And thank you again for the great suggestions! Your suggestions significantly improved this draft.

---

### Official Review · Reviewer_m3sH · 2023-07-10

**Soundness:** 3 good
**Presentation:** 3 good
**Contribution:** 3 good
**Rating:** 8
**Confidence:** 4

**Summary:**

This paper delves into the token-crisis issue in language models, a situation characterized by performance decline when the same pre-training data is used across multiple epochs. The authors scrutinize several methods of training language models with recurring tokens, encompassing regularization strategies and the use of mixture-of-experts for refined hyper-parameter tuning. Furthermore, they look into the possibilities of creating supplementary data utilizing existing language models and formulating more data-conservative model structures. The paper provides a comprehensive analysis of the token-crisis issue and its roots, as well as the efficacy of various strategies aimed at alleviating this problem.

**Strengths:**

This paper has several strengths across different dimensions:

- The paper focuses an important problem in language modeling, the token-crisis problem, which has not been extensively studied before and has big potential. The authors provide a thorough investigation of the problem and explore various approaches to mitigating it, including regularization techniques and mixture-of-experts for efficient hyper-parameter tuning.
- The presented insights have the potential to improve the performance and efficiency of language model learning. The paper's contributions, including a thorough investigation of the token-crisis problem and its causes, as well as the effectiveness of various approaches to mitigating this issue, are significant for the field of natural language processing.
- The paper is well-written and well-organized, with clear explanations of the problem and the proposed solutions. The authors provide a detailed analysis of the factors contributing to multi-epoch degradation and the effectiveness of various approaches to mitigating this issue. The experiments are well-designed and the results are presented clearly and comprehensively.

**Weaknesses:**

The first weakness of this work is that all the experiments were conducted on the T5-style masked language modeling, which is opposite to the recently surging GPT-style (i.e., causal) language modeling. And it's unclear whether all the sights/findings are applicable or transferable to the CLMs.

Meantime, the experiments were conducted on models with up to 3B parameters and did not explore the performance of the proposed approaches on larger models, such as the GPT3 or its equivalents. This limits the generalizability of the results to larger-scale models.

**Questions:**

Refer to the weakness

**Limitations:**

Yes

---

> ### Author Rebuttal · Authors · 2023-08-09
>
> > Q1: All the experiments were conducted on the T5-style masked language modeling, which is opposite to the recently surging GPT-style (i.e., causal) language modeling. And it's unclear whether all the sights/findings are applicable or transferable to the CLMs.
>
> Thank you for your insights. We totally agree that the training objective is highly important in the token-crisis problem. That is the reason why we investigate UL2[2] training objective (which is used in PaLM-2[3]) in Section 3.3. The UL2 training objective not only includes the T5-style masked language modeling but also covers causal language modeling and a more challenging masked token prediction (longer masked span or higher mask ratio), which is similar to fill-in-the-middle objective[4], an advanced training objective proposed by OpenAI and used in open-sourced models like StarCoder [5].
>
> [2] Tay, Yi, et al. "Ul2: Unifying language learning paradigms." The Eleventh International Conference on Learning Representations. 2022.
>
> [3] Anil, Rohan, et al. "Palm 2 technical report." arXiv preprint arXiv:2305.10403 (2023).
>
> [4] Bavarian, Mohammad, et al. "Efficient training of language models to fill in the middle." arXiv preprint arXiv:2207.14255 (2022).
>
> [5] Li, Raymond, et al. "StarCoder: may the source be with you!." arXiv preprint arXiv:2305.06161 (2023).
>
>
> > Q2: The experiments were conducted on models with up to 3B parameters and did not explore the performance of the proposed approaches on larger models, such as the GPT3 or its equivalents.
>
> Thank you for your valuable suggestion. We fully agree that it would have potential benefits of extending our experiments to models like GPT-3 or its counterparts. However, as mentioned in our response to Reviewer SEVe's Question 1, the primary challenge lies in resource availability. Training a 175B model on a substantial dataset for multiple epochs requires significant computational resources, which only a handful of institutions possess.
>
> It's worth emphasizing that our interest extends to conducting ablation studies that delve into the various components of multi-epoch training. These studies necessitate even more computational resources than training LLaMA-2 70B from scratch if we consider larger models. In the current paper, even focusing solely on Figure 10 comes with a considerable cost, approximately 47K USD for Google Cloud TPU usage. This cost is approximately seven times higher than training BERT-Large from scratch.
>
> To contribute to our research community, we have to scale down to align with a reasonable budget. At the same time,  to make our conclusion as sound as possible, as depicted in Figure 3 and Figure 4, we conducted experiments to verify larger model is usually more data-hungry and easier to overfit within fewer epochs.

---

### Official Review · Reviewer_SEVe · 2023-07-24

**Soundness:** 2 fair
**Presentation:** 3 good
**Contribution:** 2 fair
**Rating:** 4
**Confidence:** 4

**Summary:**

Data is one of the most important factors for training a well-performed Large Language Model(LLM). There are not enough studies that are deep into the effect of data. This paper addresses this key problem and analyzes the effect from the important aspects including the effect of pre-training, the effect of downstream tasks, and so on.

**Strengths:**

1. Data is very important for LLM. This work analyzes the impact of data from various aspects.
2. The experiments are rich and detailed.
3. The paper is well-written except some typos, for example missing ‘.’ in Line 74.


**Weaknesses:**

1. The backbone of this paper is T5 1.1. And we know that GPT is one of the most important generative models. Hence, it is necessary to add GPT as the backbone.
2. Data is the key factor for LLM. The data volume used in the experiment could affect the conclusion seriously. In this work, the maximum amount of data is 2**35, about 34B data. It is so small data for training LLM that the experiment results may not be reliable enough.
The figures of training loss over train tokens from LLAMA 1 and LLAMA 2 show that 300~400B data could display a stable trend (at least 200B). In other words, 2**27(0.13B) tokens repeated many times are different with 1TB tokens repeated the same times.


**Questions:**

1. Please update the experiment results with more data, for example, 400B.
2. If possible, show the performance of GPT-3.

**Limitations:**

Please see the weakness.

---

> ### Author Rebuttal · Authors · 2023-08-09
>
> > Q1: Use GPT instead of T5 as backbone.
>
> Thank you so much for your valuable insights. As we discussed in Appendix A.1, the differences between the encoder-decoder and decoder-only models might not be as big as our community thought. There are two main distinctions to consider: (1) The encoder-decoder architecture actually has approximately twice the number of "trainable parameters" compared to the decoder-only model. This is one important reason why encoder-deocder is sometimes better. (2) The encoder-decoder setup requires clearly separated input-output pairs to work effectively, enabling bidirectional attention only on the input tokens. Interestingly, this mechanism is also employed in decoder-only prefix-LMs. An example of such a model is UPaLM [1]. They fine-tuned PaLM (a model trained by casual LM objective) to create UPaLM with just a few training steps. This suggests that the difference between prefix-LMs and casual LMs is relatively minor. Moreover, since both prefix-LM and encoder-decoder use input-only bidirectional attention, we can view the prefix-LM as an encoder-decoder model that shares the trainable weights across the encoder and decoder.
>
> To show the encoder-decoder and decoder-only are not that different from the model architecture perspective, we conduct another set of experiments in Figure 1 in our rebuttal pdf. We can see the encoder-decoder is clearly better than vanilla decoder-only model due to more parameters. If we use MoE-based decoder-only model with comparable trainable parameters with encoder-decoder model, the gap can be patched almost perfectly.
>
> In addition, to be honest, this project was initiated before ChatGPT gained widespread popularity. As evident in our paper, we explored various factors related to the "token-crisis" phenomenon, which necessitated a wide array of implementation and timely training efforts. As a result, we opted for the well-established T5 framework, which was widely studied at the time. More importantly, once again, we believe the T5 architecture is quite similar to the widely used decoder-only design.
>
>
> > Q2: If possible, please update the experiment results with more data, for example, 400B.
>
> Thank you so much for your suggestion. We totally agree that super-scale experiments like training a 175B model with 400B tokens for multiple epochs would make our conclusion more solid. This was also mentioned in our Appendix A.3. While we acknowledge the potential benefits of super-scale experiments, resource limitations (especially during the rebuttal phase) prevented us from pursuing this scale. If we have enough computation resource, we promise that we will have a run on this level. However, considering the resource we have so far, to make our conclusion as sound as possible, as depicted in Figure 3 and Figure 4, we conducted experiments to verify larger model is usually more data-hungry and easier to overfit within fewer epochs. This supports that our insights based on the smaller model with more epochs can also apply to larger models with fewer epochs.
>
> [1] Tay, Yi, et al. "Transcending scaling laws with 0.1% extra compute." arXiv preprint arXiv:2210.11399 (2022).

---

> > ### Comment · Reviewer_SEVe · 2023-08-14
> >
> > I have read the rebuttal, thanks to the authors for their rebuttal. My score is unchanged, as most of my concerns remain:
> > 1. Both encoder-decoder and decoder-only structures are needed.
> > 2. From the loss figures of LLAMA2 (and LLAMA1), we could see clearly that 400B (at least 250B) tokens could show the tendency.

---

> > > ### Author Response · Authors · 2023-08-14
> > > **Official Comment by Authors**
> > >
> > > Thank you so much for reading our rebuttal.
> > >
> > > **1. Encoder-Decoder vs. Decoder-Only Comparison:**
> > > As shown in the Rebuttal PDF file Figure 1, we have shown that the encoder-decoder is not that different from decoder-only. Also, the real difference is the training objective, as stated in the UL2 paper above. And we took a closer look at the training objective in our paper.
> > >
> > > **2. Scale of Experiments:**
> > > While we concur on the potential advantages of conducting larger-scale experiments, we respectfully maintain our perspective that a 400B scale is not an absolute requirement for acceptance of a transformer scaling paper.
> > >
> > > - **Learning Schedule Dependency:** The training loss trend greatly depends on the learning schedule. For instance, LLaMA 1 used 4M batch size for 250K steps with a cosine LR schedule but our model used 64K batch size for 500K steps with an inverse square root LR schedule. The large batch size and cosine learning schedule of LLaMA 1 make models achieve smaller learning rate until over 100B tokens, leading to later stable loss. But our model can achieve a smaller learning rate with much fewer tokens, so that, as shown in Figure 5 of this paper, models with enough tokens can achieve stable training loss much faster.
> > >
> > > - **Addressing Cost and Accessibility:** Second, training a Billion-level model at such a scale is indeed prohibitively expensive for most institutes. There is no doubt we should scale the experiments to a moderate size. We believe that the 3B model on over 30B tokens ablation studies in this paper has been not cheap for most institutes and definitely not trivial (One single run has been around 6 times more expensive than training BERT-Large from scratch and we actually run these experiments for many times in our ablation study.). Few institutes are as rich as the big techs. We sincerely hope our community can be inclusive to invite broader participation and a wider range of insights. We believe this would not only accelerates the pace of transformer scaling research but also nurtures a diverse pool of scaling researchers worldwide.

---

### Author Rebuttal · Authors · 2023-08-09

Dear reviewers and AC:

We thank all the reviewers for their feedback. Two main concerns and concise arguments are summarized below. The full version can be found in the corresponding response of different reviewers.

* Use Decoder-only Model instead of Encoder-Decoder Model. =>  This project started before the popularity of ChatGPT. Due to the wide range of experiments, we did not finish this work until 2 months ago. As a result, we opted for the well-established T5 framework, which was widely studied at the time. More importantly, we argue that decoder-only architecture is not that different from the encoder-decoder architecture. As we discussed in Appendix A.1, there are two main distinctions to consider: (1) The encoder-decoder architecture has approximately twice the number of "trainable parameters" compared to the decoder-only model. (2) The encoder-decoder requires clearly separated input-output pairs to apply bidirectional attention only to the input tokens. To verify the viewpoint above, we compare encoder-decoder, decoder-only, and MoE-based decoder-only models in Figure 1 of our rebuttal pdf. We can see encoder-decoder is clearly better than decoder-only but the MoE based decoder-only model with comparable trainable parameters with encoder-decoder performs almost the same as encoder-decoder model. Therefore, as suggested by UL2[2] paper, the different behaviors of encoder-decoder and decoder-only are more from the training objective instead of model architecture. That is the reason why we explore UL2 training objective in our Section 3.3.

* Larger Model and More Training Data. -> As we stated in Appendix A.3, we fully agree that a super-scale investigation would be helpful, but, since we want to investigate a number of different factors resulting in token-crisis and multi-epoch degradation, we have to implement and try a range of ablation experiments. Using a very large model and dataset is prohibitively expensive for our setting. To verify larger model is usually more data-hungry and easier to overfit within fewer epochs, in Figure 3 and Figure 4, we investigate the scaling-up behaviors in multi-epoch training. This supports that our insights based on the smaller model with more epochs can also apply to larger models with fewer epochs.



Best,

Authors

---

### Decision · Program_Chairs · 2023-09-21

**Decision:**

Accept (poster)

**Comment:**

This paper studies the effect of repeating tokens while pretraining language models for more than one epoch, before finetuning on target tasks. Reviewers appreciated the relevance of the token-crisis problem this paper is focusing on, and the detailed analysis and ablations that are well-organized and clearly presented. Reviewers also acknowledge that the insights from this work can help researchers with limited resources train better language models.

The main concerns were around the use of encoder-decoder models (T5) while the more common architecture for most recent general purpose LLMs is decoder-only, experiments with "only" up to 3B parameters models, and experiments on limited pretraining and downstream datasets. All these limit the strength of the author's claims and condition them on the specific examined setting. After the discussion and the additional results provided in the rebuttal, most reviewers agreed that despite these limitations the insights from this paper overall constitute a meaningful contribution. I strongly recommend including the additional results on more downstream tasks to (especially given the focus on encoder-decoder and their common use for finetuning unlike few-shot with decoder-only). I also strongly recommend modifying the abstract and introduction to properly state the scope of this work up ahead to properly set up the reader's expectations before reading the rest of the paper.